# Mechanistic Insights and Potential Therapeutic Approaches in PolyQ Diseases via Autophagy

**DOI:** 10.3390/biomedicines11010162

**Published:** 2023-01-09

**Authors:** Mukul Jain, Nil Patil, Gholamreza Abdi, Maryam Abbasi Tarighat, Arifullah Mohammed, Muhammad Rajaei Ahmad Mohd Zain, Khang Wen Goh

**Affiliations:** 1Department of Lifesciences, Parul Institute of Applied Sciences, Parul University, Vadodara 391760, India; 2Lab 209 Cell and Developmental Biology Lab, Centre of Research for Development, Parul University, Vadodara 391760, India; 3Department of Biotechnology, Persian Gulf Research Institute, Persian Gulf University, Bushehr, 75169, Iran; 4Faculty of Nano and Bio Science and Technology, Persian Gulf University, Bushehr 75169, Iran; 5Department of Agriculture, Faculty of Agro-Based Industry, Universiti Malaysia Kelantan, Jeli 17600, Malaysia; 6Department of Orthopaedics, School of Medical Sciences, Universiti Sains Malaysia, Kubang Kerian 16150, Malaysia; 7Faculty of Data Science and Information Technology, INTI International University, Nilai 71800, Malaysia

**Keywords:** autophagy, polyQ, neurons, neurodegenerative diseases, Huntington’s diseases

## Abstract

Polyglutamine diseases are a group of congenital neurodegenerative diseases categorized with genomic abnormalities in the expansion of CAG triplet repeats in coding regions of specific disease-related genes. Protein aggregates are the toxic hallmark for polyQ diseases and initiate neuronal death. Autophagy is a catabolic process that aids in the removal of damaged organelles or toxic protein aggregates, a process required to maintain cellular homeostasis that has the potential to fight against neurodegenerative diseases, but this pathway gets affected under diseased conditions, as there is a direct impact on autophagy-related gene expression. The increase in the accumulation of autophagy vesicles reported in neurodegenerative diseases was due to an increase in autophagy or may have been due to a decrease in autophagy flux. These reports suggested that there is a contribution of autophagy in the pathology of diseases and regulation in the process of autophagy. It was demonstrated in various disease models of polyQ diseases that autophagy upregulation by using modulators can enhance the dissolution of toxic aggregates and delay disease progression. In this review, interaction of the autophagy pathway with polyQ diseases was analyzed, and a therapeutic approach with autophagy inducing drugs was established for disease pathogenesis.

## 1. Introduction

Autophagy is defined as the degradation of intracellular components within the lysosome; self-eating [1]. The main element of autophagy machinery is lysosomes containing various types of enzymes, such as cellular hydrolases, including lipases, proteases, glycosidases, and nucleotidases [2]. Acidification of organelles is achieved by the ATP-dependent proton pump, which is responsible for the functioning of the enzymes, as they need an acidic environment for their activities. An acidic environment causes the unfolding proteins such as proteases to enter and act on the material via the breaking of peptide bonds. These proteases convert proteins into di- and tripeptides, and finally into amino acids. There are three significant areas of autophagy in mammalian cells: micro-autophagy, chaperone-mediated autophagy (CMA), and macro-autophagy. There are also variants of autophagy based on cargo degradation (mitophagy) deterioration of mitochondria, (pexophagy) deterioration of peroxisomes, (lipophagy) deterioration of lipid droplets, and (aggregophagy) deterioration of aggregates [3]. Through CMA, only proteins can also be degraded, but through macro- and microautophagy, proteins and organelles can be degraded. The cargo in the cytosol undergoes degradation through macroautophagy by sequestering under a double membrane or autophagosome, which undergoes fusion with lysosome [4]. The lysosomal acidic environment activates acid hydrolase enzymes, which degrade the proteins and organelles. The discovery of lysosomes was described with macroautophagy [5]. Yeast genetic engineering showed the involvement of about 35 genes that contributed towards autophagy and are known as autophagy-related genes, ATG. Each step of macroautophagy is coordinated and regulated by autophagy proteins, which initiates the recruitment of different autophagy complexes to the internal membranes of autophagosomes [6]. The Mammalian target of rapamycin (mTOR) and target of rapamycin complex 1 (TORC1) complex negatively regulates the process of autophagy in the cell. Different phenomena (intracellular or extracellular) are recognized by ATP, mTOR, insulin, amino acids, hormones, stress factors, and glucose, which can revert by modulating the process of autophagy [7].

## 2. Mechanism of Autophagy

Autophagy is a degradation process that degrades intracellular components and has been preserved throughout evolution [8]. The autophagosomes, a double-membrane vesicle that catches cytoplasmic contents and merges with lysosomes to produce autophagolysosomes, configurations in which cargo substrates are destroyed by lysosomal enzymes, is formed during macroautophagy [9] (Figure 1).

Chaperone-mediated autophagy is defined by the direct import of cytoplasmic components into the lysosome, where they are destroyed. To digest intracellular cargo, all three kinds of autophagy depend on functioning lysosomes [10]. This complicated process may be broken down into five steps: initiation, elongation, maturation, fusion, and destruction, all of which are regulated through a series of autophagy-related genes products (ATGs) [11]. A protein kinase complex that responds to upstream cues (ATG1 and ATG13 in yeast) is involved in the commencement of the autophagy process. mTOR is a serine/threonine protein kinase that is part of the mTOR complex 1 (mTORC1), which represses autophagy in nutrient-rich environments. Enzymes involved in the synthesis of phosphatidylinositol 3-phosphate (PI3P) govern autophagosome nucleation and formation, including phosphatidylinositol 3-kinase (C3PI3K), a class III phosphatidylinositol 3-kinase (PI3K) that regulates autophagy by mediating the recruitment of other autophagy-regulatory proteins to the pre-autophagosomal structure. B-cell lymphoma-2 (BCL-2), interacting protein (Beclin1), and other vital components, including VPS34 (C3PI3K), make up part of the nucleation complex. At the endoplasmic reticulum (ER) membrane, the anti-apoptotic proteins BCL-2 and BCL-XL suppress Beclin 1. ATG5/ATG7-independent autophagy has also been linked to Beclin 1 [12]. ATG2A, or ATG2B, WD repeat domain 45 (WDR45)/WD40 repeat protein interacting with phosphoinositide 4 (WIPI4), and the transmembrane protein ATG9A, form the ATG9 trafficking system, which provides a membrane and assists in the elongation of the phagophore, the initial sequestering compartment. Two ubiquitin-like (Ubl) conjugation systems, involving the Ubl proteins ATG12 and ATG8-family proteins, are also involved in membrane expansion (including the microtubule-associated protein 1 light chain 3 (LC3) and gamma aminobutyric acid receptor-associated protein (GABARAP) subfamilies). ATG12 binds ATG16L1 after being conjugated to ATG5 by the E1 and E2 enzymes ATG7 and ATG10, respectively. Phosphatidylethanolamine (PE), a membrane-resident lipid, is coupled to ATG8-family proteins. The protease ATG4, the E1 enzyme ATG7, the E2 enzyme ATG3, and the E3 enzyme ATG12–ATG5–ATG16L1 complex must all work together to attach PE [13]. The cargo sequestration is accomplished throughout the maturation phase, and the autophagosomes merge with lysosomes to destroy their cargo. Important components in autophagosomes’ maturation are SNARE (soluble N-ethylmaleimide-sensitive fusion protein receptor) proteins, the endosomal coat protein complex (COPs), the endosomal sorting complexes required for transport (ESCRT-III) complex, small GTPase-Rab proteins, members of the chaperone heat shock protein 70 (HSP70) family, and tyrosine kinase protein 1 (TECPR1) [14]. The cargos are digested in the autolysosomes by lysosomal hydrolase, and the productions are delivered back to the cytosol by lysosomal permease in the last stage. Those hydrolyzed proteins interact with vacuoles exhibiting autophagosomal and auto lysosomal characteristics in a polyQ-length-dependent manner; that process attempts to establish a relationship between autophagy and neurodegenerative disorders (polyQ diseases) [15].

## 3. PolyQ Diseases, Their Pathophysiology, and Therapeutics

Polyglutamine diseases are a group of hereditary neurodegenerative diseases categorized with genomic abnormalities in the expression of cytosine–adenine–guanine (CAG) triplet repeats in coding regions of specific genes related to disease [16]. The CAG triplet codon codes for glutamine (one-letter code, Q), and its expansion in the disease-causative genes causes the formation of amyloid proteins with a peculiar pathogenic extension of the polyQ tract. Fischbeck and coworkers in 1991 for the first time observed the extension of CAG repeats in exon 1 of the androgen receptor gene of spinal and bulbar muscular atrophy (SBMA)-affected patients, after which it has been discovered in other different inherited neurodegenerative problems [17]. A common characteristic of polyQ ailments is the revolutionary deterioration of neurons in the specific regions of the brain, which cause impairment in basic functioning, such as motor disturbances that depend on the part of the brain that is affected. The proteasome and lysosome are unable to break down compact complexes made by these expanded polyQ tracts, which can lead to abnormalities in various intracellular pathways and dysfunction in the neurons. Theoretically, an expanded polyQ tract can result in protein misfolding and the formation of insoluble aggregates, which then gather inside neurons as inclusion bodies. The expanded polyQ tract inside a target protein facilitates the harmful aggregate-prone conformational change. It has been established that if polyQ is at least 37 residues long, it will spontaneously fold into a β-helix structure. Perturbation of normal protein function, transcriptional dysregulation, and organelle dysfunction may induce polyQ toxicity mutation. There are so far nine disorders in this group of diseases, which include Huntington’s ailment (HD); spinocerebellar ataxia (SCA) types 1, 2, 3, 6, 7, and 17; and Dentatorubral pallidoluysian atrophy (DRPLA) [18]. Since misfolding and aggregation of the expanded polyQ tract are assumed to be the first events in the common pathogenic cascade of the polyQ diseases, suppression of the polyQ aggregation is expected to have a wide-ranging impact on the functional abnormalities of many downstream cellular processes that are impacted by the aggregation process of the polyQ proteins [19]. Considering these benefits, inhibition of polyQ protein misfolding and aggregation has received substantial research, being the optimal therapeutic strategy for the creation of disease-modifying treatments for polyQ illnesses [20]. In the subsections that follow, we will concentrate on two strategies for preventing polyQ protein aggregation. One strategy is to create powerful inhibitors, such as small chemical compounds and short peptides, which are intended or chosen to bind specifically to the expanded polyQ tract and prevent polyQ protein aggregation. The alternative strategy is to engage cellular defense mechanisms that stop the formation of aggregates and buildup of misfolded proteins. Both methods effectively slow down the aggregation of enlarged polyQ proteins and the courses of illness and animal model traits, demonstrating the efficacy of this therapeutic target for drug development [21].

## 4. Autophagy Dysfunction in Polyglutamine Disease

There are two degradation systems reported under the protein quality control system to get rid of these misfolded proteins—the ubiquitin-proteasome system (UPS) or autophagy [22]. In the eukaryotic system, these long polyQ stretches structurally transform into misfolded proteins, which cannot be degraded by UPS machinery, as there is the sequestration of UPS components, due to which delivery of misfolded proteins in polyQ disorders cannot reach out to the nuclear proteasomes [23]. Due to increases in UPS components and heat shock proteins in nuclear inclusions, these systems are deficient, which causes aggravation of polyQ aggregates that become the substrates for autophagy degradation. The autophagy process gets upregulated to counter the protein aggregates and maintain protein homeostasis. The autophagy process is limited to the cytosol, unlike the UPS, which is active in both the nucleus and the cytosol. Degradation of these aggregates in polyQ disorders by autophagy is directly correlated with its localization (Figure 2). As autophagy machinery is activated only in the cytosol, the aggregates present in the cytosol can be maintained, but in certain diseases, formation of aggregates takes place in the nucleus, as in SCA1, SCA7, SCA17, and SBMA, where it becomes problematic. In HD, SCA3 and DRPLA aggregates are distributed in both the cytoplasm and the nucleus [24]. These reports suggest that the aggregates formed in the cytosol can be degraded by autophagy, and autophagy upregulation degrades them and prevents their uptake inside the nucleus. The nucleus plays a central role in the pathogenesis in a polyglutamine diseases. Further on, detailed interaction between autophagy and various polyQ diseases is elucidated.

### 4.1. Huntington’s Disease

Among all nine polyQ-associated neurodegenerative diseases, the most frequent autosomal dominant disease is Huntington’s disease (HD), in which there is an expansion of CAG repeats (≥36) [25]. Most HD patients have a functioning copy of wild type HTT (wtHTT) because they are heterozygous for the HTT mutation (mHTT). HTT plays a crucial role in maintaining the health of neurons and is crucial for embryonic development. Adult-onset loss of complete HTT causes neurodegeneration. While there have been several attempts to comprehend and create novel treatments that can support pathways that can remove mHTT and its aggregates, it might be critical to do so selectively without affecting wtHTT levels [26]. This disease is characterized by involuntary motor movements, cognitive decline, and psychiatric illness. There are increases in the expression of autophagy markers p62 (a cargo-adaptor) and LC3-II in the brains of HD transgenic mice (an autophagosome membrane marker) [27].

Defects in autophagy machinery and in ubiquitin-proteasome system (UPS) are the hallmarks of many neurodegenerative diseases, including HD [28]. The polyglutamine tracts causing various toxicities are assumed to hinder the autophagy-related functions of wild-type protein in HD [29]. Similarly, the presence of aggregates of mutated N- terminal Htt protein or inclusion bodies (IB) in the cell nucleus depicts the alterations in UPS which play major roles in the removal of toxic protein aggregates, specifically in the nucleus [30]. Various lines of evidence suggest that the pathogenesis of HD is the outcome of defects in autophagy machinery and loss of normal protein function in this cleaning crew and in the UPS system. The formation of autophagosomes and their clearance are normal in HD, but these autophagosomes are usually empty due to failure in cargo trapping. The pathophysiology of mutation in several autophagy-related genes is similar to the pathological findings associated with HD. Studies on loss of wild-type protein functions in mouse and *Drosophila* models revealed several characteristics of disrupted autophagy. Both wild-type protein in association with HAP1 and the mutated one can alter the axonal trafficking of autophagosomes and vesicles in neurons in vitro and in vivo [31]. In *Drosophila*, the loss of wild-type protein during embryogenesis results in viable embryos, but later in adult life, absence of it leads to several neurodegenerative symptoms, reduced viability, mobility, reduced or collapsed thorax, etc. [32]. Defects in starvation-induced autophagy have also been reported in Drosophila larvae and Htt conditional knockout of mouse CNS, resulting in accumulation of protein sequestosome 1 (p62/SQSTM) in striatum over time, a hallmark of disturbed autophagy. A master regulator of autophagy, mTOR (mammalian target of rapamycin), is a serine-threonine kinase, and it is sequestered with polyQ aggregates, which cause dysfunction in its kinase activity and induce autophagy in HD. PolyQ-Htt aggregates impaired the formation of autophagosomes by requisite with Beclin-1 (an autophagy initiation protein) [33]. HD is an age-onset disorder, as the lysosomal clearance and autophagy activity decrease with age, and the aggregates get exacerbated in the progression of HD [34].

### 4.2. Spinal and Bulbar Muscular Atrophy

It is also known as Kennedy disease and is an X-linked inherited neuromuscular disorder. It leads to weakness and bulbar muscular atrophy [35]. Pathways that control cellular proteostasis have drawn a lot of interest since the unfolded mutant protein is the primary mediator of toxicity in SBMA. One of them is macroautophagy, often known as “autophagy”, a highly conserved catabolic process in which misfolded or defective proteins and organelles in the cytoplasm are encapsulated in double-membraned structures called “autophagosomes” [36]. These autophagosomes are transported to lysosomes and fused with them, allowing their intraluminal contents to degrade. The coordinated lysosomal expression and regulation (CLEAR) network, which is one of the mechanisms used to regulate autophagy, includes the transcription factor EB (TFEB), which controls the expression of hundreds of genes relevant to lysosomes and autophagy [37]. The main regulator of both autophagy and lysosomal biogenesis is TFEB. Aside from promoting mTOR inhibition and dietary restriction, therapies that cause autophagy also increase TFEB activity [38]. Although there is evidence that SBMA alters the regulation of protein quality, it is unclear how these regulatory transcription factors are impacted by illness [39]. In contrast to the range of 8 to 34 polyQ stretches, there is an accumulation of LC3 puncta in polyglutamine androgen receptor (polyQ-AR) disease [40]. It is reported that there is an accumulation of AV and multivesicular body in fly model of AR, which causes degeneration of the eye [41]. In the mice model systems of AR, an increase in autophagy vacuoles at later stages suggests an alteration in autophagy [42].

### 4.3. Spinocerebellar Ataxia Type 1

Cerebellar ataxia type 1 is a late-onset autosomal dominant neurodegenerative condition characterized by cerebellar ataxia, oculomotor abnormalities, pyramidal and extrapyramidal characteristics, peripheral neuropathy, and cognitive impairment to varying degrees [43]. Ataxin 1 (ATXN1), the protein that causes spinocerebellar ataxia type 1, has an ataxin-1 and HMG-box protein 1 (AXH) sequence, which was initially discovered in the high-mobility group transcription activator HMG1, gave the first evidence tying ATXN1 to transcription. The AXH domain has been found to interact with at least one trans-activator, SP1, to control transcription directly [44]. SCA1 is defined by a gradual loss of motor coordination, which generally begins with a shaky stride and balance and cognitive difficulties. The mutant ATXN1 protein promotes SCA1 degeneration in cerebellar Purkinje cells and neurons in the brainstem and spinal cord. Overexpression of the chaperones has been proven to slow the course of illness. One of the most common cellular chaperone proteins is heat shock protein 90 (Hsp90). A recent study has found that Hsp90 inhibitors reduce misfolded protein aggregation and toxicity by affecting the autophagy-lysosome pathway (ALP) [45].

### 4.4. Spinocerebellar Ataxia Type 2

Spinocerebellar ataxia type 2 (SCA2) is the most common polyQ ataxia in Cuba, India, Mexico, and southern Italy, and it has a vast geographical spread. The length of the CAG repeat expansion determines the age of onset in SCA2 and other polyQ diseases: the longer the polyQ expansion, the earlier the onset of disease. In SCA2, the average age at the beginning of symptoms is around 35 years [46]. The autophagic protein WD repeat and FYVE domain containing 3 protein (WDFY3) was shown to be overexpressed in patients with SCA2. Due to autophagy malfunction, the STAU1 (staufen1) protein was elevated in SCA2 patients’ fibroblasts [47]. When ubiquitin-labelled proteins amass and assemble, a selective lysosomal-mediated degradation mechanism, autophagy, destroys them. This mechanism requires the presence of microtubule-associated protein 1 light-chain 3 (LC3), p62, and autophagy-linked FYVE domain protein (Alfy). LC3 (gene MAP1LC3) is an autophagy adaptor protein that is necessary for autophagosome formation and maturation. P62 (gene SQSTM1) connects ubiquitinated cargo to the nascent autophagosomes, allowing it to mature and fuse with lysosomes. Alfy (gene WDFY3) is an autophagy scaffold protein that binds p62 and is essential for the autophagic degradation of cytoplasmic ubiquitin-positive inclusions [48].

### 4.5. Spinocerebellar Ataxia Type 3

Spinocerebellar ataxia type 3 (SCA3) is also known as Machado–Joseph disease or MJD [49]. Progressive ataxia, peripheral amyotrophy, muscular atrophy, parkinsonian symptoms, dystonia, and spasticity are the predominant clinical characteristics [50]. Major clinical symptoms include dysarthria, eyelid retraction, ocular motility abnormalities, and dystonia. PolyQ-ATXN3 in soluble form can be degraded by autophagy induction [51].

A number of stresses, including as the presence of cytotoxic protein aggregates, can cause autophagy, which controls nutrient recycling during times of nutritional scarcity and ensures cell survival [52]. Phosphatidylethanolamine (PE) conjugation and autophagosomal anchoring of six closely related ubiquitin-like modifiers from the Atg8 family—LC3A, LC3B, LC3C, GABARAP, GABARAPL1, and GABARAPL2—control the beginning, development, and elongation of the autophagosomal membrane [53]. By directly engaging with these proteins through conserved LC3-interacting areas, LC3s and GABARAPs play key roles in aiding the recruitment of cargo receptors and other autophagy proteins to autophagosomes (LIRs) [54]. The portion of LC3/GABARAP located on the luminal side of autophagic vesicles is destroyed in the lysosomes together with cargo receptors and substrates, whereas LC3/GABARAP located on the cytosolic face is released by Atg4 protease after fusion with lysosomes [55]. It was observed that a defect in autophagy increased retinal toxicity in the degeneration model of Drosophila polyQ-ATXN3 [56]. Interestingly, numerous members of the heat-shock protein (Hsp) chaperone family were discovered to be dependent on intact autophagy function and were identified as genetic modifiers of polyQ-ATXN3 toxicity in an eye degeneration model [57]. In the Drosophila model, genetically inhibiting autophagy can greatly worsen polyQ-neurotoxicity ataxin-3 on retinal cells, pointing to autophagy’s protective function in SCA3. By blocking mTOR, the autophagy inducer rapamycin can increase autophagy’s activity and encourage the breakdown of ataxin-3 aggregation [58].

### 4.6. Spinocerebellar Ataxia Type 7

In 1995, the gene responsible for SCA7 was discovered on chromosome 3p12-21.1. The ATXN7 gene was discovered two years later after library screening for CAG repeat sequences. The ATXN7 gene is made up of 13 exons that span 140 kb of genomic DNA and encodes a polymorphic polyQ stretch in the amino terminus. CAG expansions are a dynamic type of mutation that can occur in both germline and somatic tissues. SCA7 CAG repeats have the most potential to increase the following transmission among CAG/polyQ diseases [59]. The adult-onset type is caused by the majority of SCA7 alleles, which have 36–55 CAG repeats and proceed over several decades until death [60]. ATXN7 is a component of the transcriptional co-activator complex SPT-TAF-ADA-GCN5 acetyltransferase (STAGA), which remodels chromatin and exhibits histone acetyltransferase activity. Only a few studies have looked at the function of autophagy dysregulation in SCA7 before recently. However, studies from other polyQ illnesses suggested that polyQ-ATXN7 toxicity might target autophagy. In the context of SCA-7, ATXN7 is a recognized component of the multiprotein SAGA complex, a transcriptional coactivator that has been conserved from yeast to humans. This transcriptional activator (RNAPII) depends on SAGA’s chromatin modelling activity, which directly correlates with the lysosomal membrane degradation (microautophagy) [61]. Proteasome components, protein chaperones, and ubiquitin were found in nuclear inclusions in SCA7 patient brains. When polyQ-ATXN7 is overexpressed in cells, the number of autophagic vesicles increases (AVs) [62].

### 4.7. Spinocerebellar Ataxia Type 17

SCA17 is a neurodegenerative illness caused by the extension of a CAG/CAA repeat in the TATA box-binding protein (TBP) gene’s coding region beyond 43–45 units. That TBP interacts with other protein factors, including high mobility group box 1 (HMGB1), to regulate gene expression [63]. HMGB1 was found to be sequestered into polyQ-TBP aggregates in vitro, leading to impaired starvation-induced autophagy. HMGB1 is an important autophagy regulator, raising the prospect of a general mechanism for autophagy dysfunction in all polyQ diseases via HMGB1 inhibition [64]. SCA17 has a complicated and changeable clinical phenotype that, in some situations, resembles Huntington’s disease; as a result, it is also known as Huntington’s disease-like 4 (HDL-4). The disease usually appears in the third or fourth decade, with the presenting symptoms being either mental problems such as behavioral changes and mood swings or neurological characteristics such as ataxia and mobility difficulties. A common early complaint is cognitive impairment (77%) [65]. Chaperones are known to protect neurons against cellular stress and acute and chronic stresses that occur in neuronal cells in the brain as people age [66]. Reduced brain chaperone concentrations presumably reduce cellular capacity to refold polyglutamine proteins and protect against oxidative stress as people age, causing age-related neuropathology. Although it has been claimed that increasing chaperone activity in brain cells might minimize misfolded protein toxicity, the complexity of chaperones has made pinpointing precise targets for successful therapy problematic. This emphasizes the need to identify possible treatment targets by decoding the transcriptionally dysregulated pathways in SCAs [67].

### 4.8. Dentatorubral Pallidoluysian Atrophy (DRPLA)

The amplification of a CAG stretches in the atrophin-1 gene causes dentatorubral–pallidoluysian atrophy (DRPLA), a human polyQ ataxia [68]. Atrophin-1 (ATN1) is a bimodal transcriptional cofactor that is attracted to regulatory regions by a variety of transcription factors. It is involved in various physiological activities. A method called nucleophagy is triggered to damage abnormal nuclear constituents by macroautophagy. By binding to autophagy adaptor proteins such as p62/SQSTM1, certain cargos are prepared for destruction. The lipidated version of microtubule-associated protein 1 light chain 3b (LC3B-II) is likewise bound by these adaptors. An autophagosome is a vesicle with a protein attached in its growing double membrane [69]. After vesicle occlusion, autophagosomes continue to develop and merge with lysosomes to form autolysosomes, where the cargo is broken down and recycled by lysosomal hydrolases. Autophagy controls axonal homeostasis and eliminates cytoplasmic aggregation-prone proteins, which reduces the etiology and progression of neurodegenerative illness. Nucleophagy is a kind of autophagy linked with the destruction of lamin B1, a constituent of the nuclear lamina, by direct contact with LC3B-II [70]. The fruit fly *Drosophila melanogaster* is one of the invertebrate models for DRPLA neurodegeneration. When a human disease allele is introduced, it causes age-dependent neuronal death that may be brought on by issues with lysosomal autophagy and accumulation of the poly-Q repeat protein [71]. Autophagy is a process in which cytosol and organelles are encapsulated in double-membrane vesicles that convey the contents to the lysosome/vacuole for breakdown and recycling of the resultant molecules, which occurs in all eukaryotic cells. It normally occurs long-lived macromolecules and organelles to maintain homeostasis, differentiation, and tissue remodeling [72].

## 5. Intracellular Quality Control of Disordered Proteins by Autophagy

Autophagy, which is defined as one of the main systems for protein quality control, gets compromised under various conformation disorders [73]. In protein conformation disorders such as Parkinson’s disease or Huntington’s disease, it upregulates autophagy [74]. Upsurge in autophagy can be related as protection as the viability of cells gets compromised. Autophagy is an effective intervention in these diseases, as it degrades the aggregates and regulates the toxicity caused by aggregates. It was reported in other model systems, such as mice, flies, and worms of the HD model, that upregulating autophagy by chemical modulators decreased the proteotoxicity caused by aggregates and increased cell viability [75]. It was also suggested that autophagy is compromised and causes different protein aggregate disorders such as Parkinson’s disease, Huntington’s disease, Alzheimer’s disease, polyQ diseases, amyotrophic lateral sclerosis, and prion diseases. In different diseases, defects in the autophagy pathway are increased due to either dysfunction in the formation of autophagosomes, selective identification of cargo, mobilization of autophagosome, or fusion of lysosome with autophagosome [76]. Defects in autophagy components, such as mutations in presenilin I, which cause defects in the acidification of lysosomes, induce autophagy defects, and finally cause Alzheimer’s disease [77]. It was also reported that there is initial upregulation of autophagy, which was due to the compensation in UPS and CMA. Defects in CMA were also reported as the causes of neurodegenerative disorders. α-synuclein, whose accumulations are observed in Parkinson’s disease (PD), degraded through CMA. Rapamycin is an activator of macroautophagy, and it is a beneficial chemical modulator for the degradation of protein aggregates. As mTOR controls various other cell processes, it cannot be used for clinical applicability. Autophagy induction independent of mTOR would be beneficial for the suppression of aggregates, and lithium can be used as the target for the stimulatory effect of autophagy [78]. The limitation of these modulators is that they act on the initial phase of autophagy, i.e., formation of autophagosomes. No effect was seen on diseases, which show defects in the late phase of autophagy.

## 6. Autophagy Modulators as Therapeutics against polyQ Diseases

Intracellular protein aggregates arise when the amount of misfolded proteins exceeds a particular level, and they are subsequently systematized into small soluble oligomers. With time and rising protein concentrations, oligomers degrade into more metabolically stable insoluble clumps. As this oligomerization is prolonged, it will take a long time. As a result, protein misfolding might begin years before the formation of protein aggregates [79]. This autophagy deficiency has been connected to neurodegeneration and ageing. Autophagy failure leads to the accumulating of several hazardous aggregate-prone proteins that cause neurodegenerative disorders, such as mutant huntingtin, alpha-synuclein, tau, and others [80]. This autophagy dysfunction causes different disorders, and these disorders are treated by different types of autophagy modulators. Rapamycin, CCI-779, rilmenidine, lithium, liraglutide, neferine, and histone deacetylase inhibitors (HDIs) such as autophagy modulators are used as therapeutics against HD, which have various mechanisms, such as iInhibition and activation of mTORC1 and AMPK. Histone deacetylase inhibitor causes DNA methylation modifications in Huntington’s disease (HD). These modulators directly or indirectly cause deactivation or regulation of the process of autophagy [29]. Autophagy modulators, which function on distinct autophagy pathways, are the subject of research. These modulators have the potential to act on a specific pathway point which is shown in Figure 3. There are several routes, each with its own set of modulators, such as AMPK, mTORC1, histone deacetylase (HDAC), Unc-51-like kinase 1 (ULK1), and BECN1. Rapamycin, which inhibits mTOR, improves cognitive and memory deficits, reduces the levels of hyperhosphorylated tau, and reduces neuronal TBPH aggregates in AD mice and Drosophila. Rapamycin also causes ageing in mice and life extension in Drosophila [81]. Trehalose activate AMPK, which induces tau clearance and neuronal survival, and enhances the degradation of mHTT aggregates in HD, ALS, and AD [50]. Lithium reduces tau phosphorylation, reduces the formation of Aβ plaques, and induces the removal of irregular mHTT aggregates [82]. Metformin, bosutinib, and nilotinib activate AMPK activation, which prevents the degeneration of dopaminergic neurons and reduces levels of phosphorylated α-syn Ser129 in Parkinson’s disease and Alzheimer’s disease. In the AMPK (adenylate-activated protein kinase) signaling pathway, the Neferine modulator inhibits activation of AMPK-alpha-1, which causes brain atrophy, increases formation of Htt aggregates, and facilitates neuronal loss [83]. In the case of mTORC1 (mammalian target of rapamycin complex 1), nutrients, neurotrophic factors, and neurotransmitters trigger the mTOR-signaling pathway, which increases protein (and perhaps lipid) production while suppressing autophagy. During development, these mechanisms promote neuronal differentiation, neurite elongation, and synapse formation, all of which contribute to proper neuronal growth. As a result, mTOR-signaling disruption may result in neuronal degeneration and improper neural development. For preventing them, rapamycin and the lithium-like autophagy modulators become helpful [84]. In the ULK1 signaling pathway, rilmenidine modulation inhibits the activation of autophagy. ULK1 is an essential mediator of type-I (interferon receptor) IFNR-generated signals that regulate gene transcription and promote anticancer responses [85]. As per the current research work, here in Table 1, various autophagy modulators are described with their modes of action, in particular, polyQ diseases. SCA1 and SCA6 exhibit remarkable phenotypic anticipation and diversity, which mostly reflects variations in repeat size across afflicted individuals. The majority of people with SCA1 show indications of extensive brainstem and cerebellar dysfunction with only modest supratentorial involvement. Acetazolamide majorly inhibits the production of bicarbonate ions and dysregulates the pH of the blood in brainstem [86]. In other polyQ diseases such as SBMA, the male hormones testosterone and dihydrotestosterone (DHT) attach to this dysfunctional receptor, harming the muscle-innervating nerve cells and resulting in weakness. DHT synthesis is reduced by dutasteride [87].

## 7. Discussion

To enable the acute manipulation of autophagy process for cell biological and physiological research, more selective autophagy modulators must be developed, both for therapeutic uses and for use as chemical probes. Several gene therapy strategies targeting PolyQ-SCAs have been used in preclinical models, such as non-human primates, patient cells, and rodent models. In the animal models, the administration of gene therapy showed significant improvements in neuropathology and disease-associated phenotypes, and a reduction in disease-associated toxicity [103]. The purposes of such strategies are to either respond to autophagy-signaling system abnormalities, enhance neuroprotection, quiet CAG-expanded transcripts, or rectify the pathogenic mutation. By attaching to ubiquitin-associated bacteria and/or viral nucleocapsids and the autophagosome-associated protein LC3 LIR domain, adapter molecules such as NBR1, p62, nuclear dot protein 52 (NDP52), and optineurin target microorganisms for autophagy. Phosphorylation regulates the action of several microbial adapter proteins; for instance, TANK-binding kinase 1 phosphorylates optineurin, boosting its LC3-binding affinity and cytosolic *S. enterica* autophagy clearance [104]. In contrast to spinal muscular atrophy, UniQure Biopharma is developing AAV5-miHTT (AMT-130), and the University of Massachusetts Medical School is developing AAVrh10 miR-SOD1 injection; both are in the first phase of clinical trials, according to May’21 data. The Avexis Pharmaceuticals has completed clinical studies for scAAV9-SMN (AVSX-101), a medication to treat spinal muscular atrophy [96]. Huntington’s disease post-mortem investigations reveal that practically full degradation of the dopamine system occurs very early in the disease’s clinical phase. Even if neuroprotective medication is successful, it may be of little help if it is started after the neurodegenerative process has progressed mostly or entirely [105]. Recent advances in the detection of mutant huntingtin (mHTT) in CSF could also help early studies be conducted. In this context, it will be crucial to investigate how changes in the huntingtin metabolism and brain accumulation in HD patients and animal models are reflected by mHTT levels in CSF or peripheral probes and whether therapeutic injection affects these levels [106,107].

## 8. Conclusions

Among the different types of cell death pathways, autophagy is the one which maintain homeostasis of cell (neurons) and dissolves misfolded aggregates consisting of β sheets. In this review, the process of autophagy pathway was elucidated in reference to various canonical and non-canonical pathways, along with how the regulation will be performed by using various modulators. Clinical trials are also going on for various autophagy modulators in relation to polyQ diseases. More understanding of autophagy pathways would lead to multi-target approaches against toxic proteins and neuronal degeneration, and delay the diseases progression.

## Figures and Tables

**Figure 1 biomedicines-11-00162-f001:**
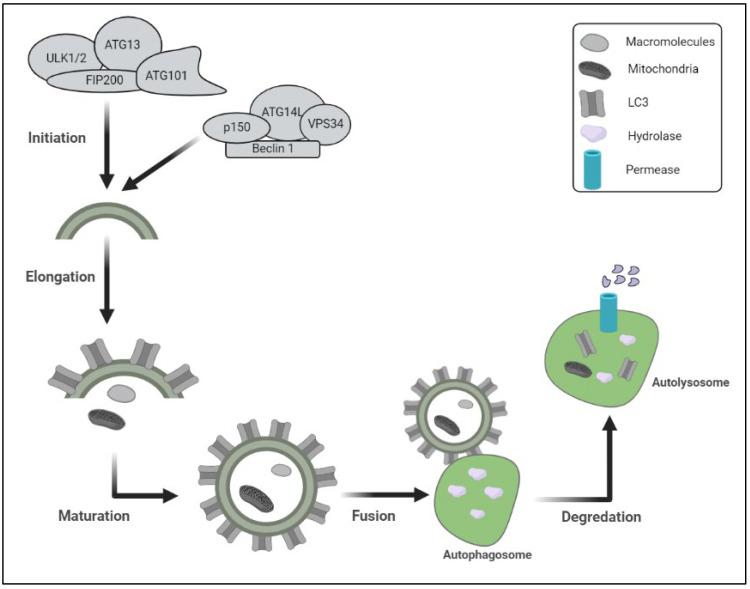
Schematic representation of the protein degradation lysosomal pathway includes different steps of autophagy and proteins involved in initiation of autophagy. *(ULK1/2: unc-51-like kinase 1; FIP200: focal adhesion kinase family-Interacting Protein of 200 kDa); ATG13: autophagy-related protein 13; p150: PI3-kinase 150; VPS34: vacuolar sorting protein-34).*

**Figure 2 biomedicines-11-00162-f002:**
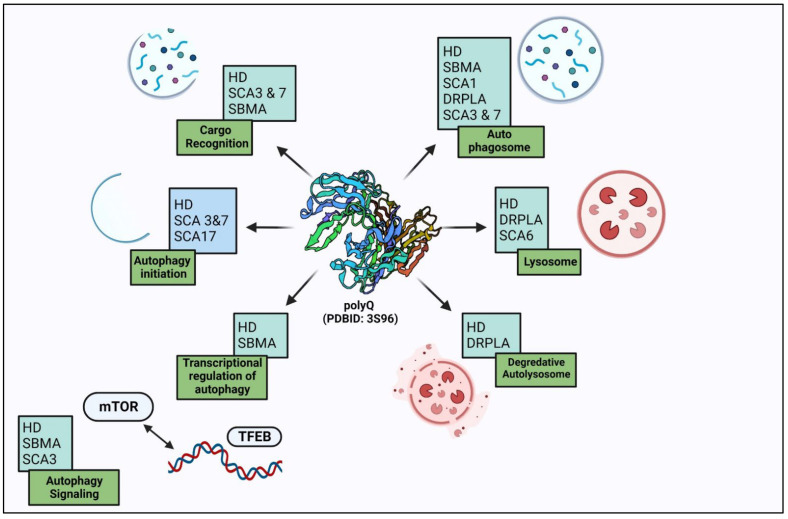
Autophagy dysregulation in polyglutamine disorders: autophagy degrades polyglutamine-expanded protein aggregates and also regulates them at different steps.

**Figure 3 biomedicines-11-00162-f003:**
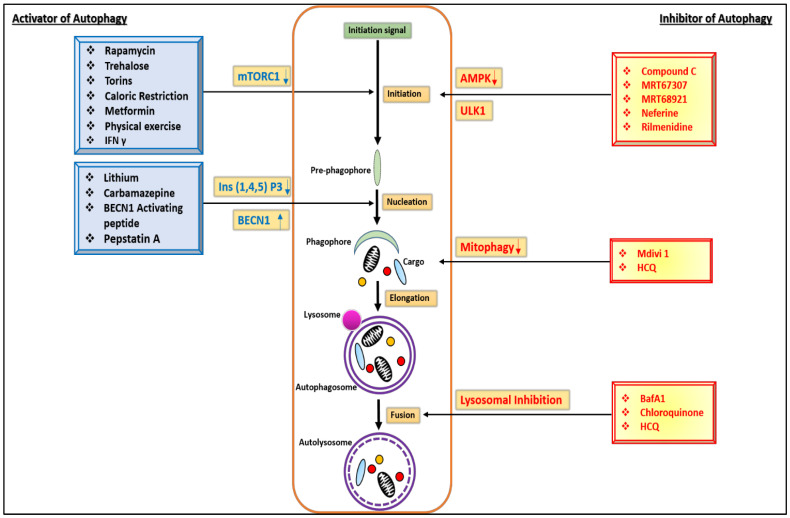
Pharmacological modulators of autophagy including activators and inhibitors at different steps of autophagy.

**Table 1 biomedicines-11-00162-t001:** Therapeutic effects and modes of action of autophagy modulators in relation to polyQ diseases.

Different Types of polyQ Diseases	Autophagy Modulators	Mode of Action	Ref.
HD	Rapamycin, CCI-779	mTORC1 inhibition	[29]
	Rilmenidine	AMPK activation	[29]
	Lithium	mTORC1 inhibition	[88]
	Trehalose	mTORC1 inhibition	[88]
	CTEP	Formation of autolysosome	[89]
	Liraglutide	AMPK activation	[90]
	Neferine	AMPK activation	[90]
	Histone Deacetylase Inhibitor	HDAC modulation	[91]
SBMA	Dutasteride	Decreases DHT production	[92]
	Trehalose	Androgen receptor (AR) degradation; Enhance autophagy	[92]
	Placebo	In primary stage decrease DHT production	[92]
DRPLA	Clonazepam	Modulation of GABA function in the brain	[93]
	Chlordiazepoxide	Increased binding of the inhibitory neurotransmitter	[93]
	Sodium valproate	Mediated through effects on the function of brain	[93]
	Piracetam	Producing a lowering of cerebral artery tonus	[94]
SCA1	Baclofen	It improves Cerebellar Impairment	[95]
	Acetazolamide	lower blood pH and carbonic anhydrase inhibitor	[96]
SCA2	Riluzole	Inhibits the release of glutamic acid from cultured neurons	[97]
SCA3	Tnezumab	Inhibiting the binding of NGF	[98]
	Fasinumab	Inhibiting the binding of NGF	[98]
SCA6	Acetazolamide	lower blood pH and carbonic anhydrase inhibitor	[96]
	Riluzole	Inhibits the release of glutamic acid from cultured neurons	[97]
SCA7	Amantadine	Increases dopamine release	[99]
	Buspirone	Prevents dopamine reuptake	[100]
	Riluzole	Inhibits the release of glutamic acid from cultured neurons	[94]
SCA17	Sodium valproate	Mediated through effects on the function of brain	[94]
	Carbamazepine	It reduces the release of a neurotransmitter—Glutamate	[98]
	Lamotrigine	Suppressing the release of the excitatory glutamate	[101]
	Topiramate	Control brain activity	[102]

## Data Availability

Not applicable.

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
