# Peer review of "Mechanistic Insights and Potential Therapeutic Approaches in PolyQ Diseases via Autophagy"

_biomedicines, 2023, doi:10.3390/biomedicines11010162_

Round 1

Reviewer 1 Report

In the present review, the Authors describe the involvement of Autophagy in the polyQ diseases and the possible therapeutic approaches to stimulate this degradative pathway for the cure of these diseases.

I have several comments about the present review.

Major Comments:

1) there is no general description of polyQ diseases

2) In the paragraph 2 "the autophagy dysfunction in polyglutamine diseases" need to be improved.

2.1) The autophagy in HD, SBMA and <SCA-3 might be studied in deep.

2.2)  In SCA-7,SCA-17 and DRPLA no description of autophagy has been detailed in the text. Is not autophagy involved in these diseases? Please, describe the autophagy impairment (if observed) in these diseases.

3) In the paragraph 5 "Autophagy modulators as therapeutics against polyQ diseases" the authors described in the text only compounds found effective in HD, but they don't describe compounds for the other polyQ diseases (e.g. trehalose). 

Moreover, they list compounds in the table that were not described in the text. Please describe them in the text. 

4) in the text is cited the Protein Quality control system, but it is not described. 

Minor Comments:

1) the paragraph 3 might be moved before the paragraph 2.

2) the english form need to be revised

Author Response

Response to Reviewer 1

  •  There is no general description of polyQ diseases

Response:

In response to the reviewer comment we have added chapter 3 “PolyQ diseases, its pathophysiology and therapeutics”

Polyglutamine diseases are an assembly of hereditary neurodegenerative diseases categorized with genomic abnormalities in the expression of cytosine-adenine-guanine (CAG) triplet repeats in coding regions of specific genes related to disease [16]. The CAG triplet codon codes for glutamine (one-letter code, Q) and its expansion in the disease-causative genes cause the formation of amyloid proteins with a peculiar pathogenic extension of polyQ tract. Fischbeck and coworkers in 1991 for the first time observed the extension of CAG repeats in exon 1 of the androgen receptor gene of spinal and bulbar muscular atrophy (SBMA) affected patients, after which it has been discovered in other different inherited neurodegenerative problems [17]. A common characteristic of polyQ ailments is the revolutionary deterioration of neurons in the specific regions of the brain, which cause impairment in basic functioning like motor disturbance that depends on the part of the brain that is affected. The proteasome and lysosome are unable to break down compact complexes made by these expanded polyQ tracts, which can lead to abnormalities in various intracellular pathways and dysfunction in the neurons. Theoretically, an expanded polyQ tract can result in protein misfolding and the formation of insoluble aggregates, which then gather inside neurons as inclusion bodies. The expanded polyQ tract inside a target protein facilitates the harmful aggregate-prone conformational change. It has been established that if polyQ is at least 37 residues long, it will spontaneously fold into a β -helix structure. Perturbation of normal protein function Transcriptional dysregulation and organelle dysfunction may induce polyQ toxicity mutation. There are so far, nine disorders mentioned that made a group of these types of diseases, which include Huntington’s ailment (HD), spinocerebellar ataxia (SCA) types 1, 2, 3, 6, 7 and 17 and Dentatorubral pallidoluysian atrophy (DRPLA) [18]. Since misfolding and aggregation of the expanded polyQ tract are assumed to be the first events in the common pathogenic cascade of the polyQ diseases, suppression of the polyQ aggregation is expected to have a wide ranging impact on the functional abnormalities of many downstream cellular processes that are impacted by the aggregation process of the polyQ proteins [19]. Considering these benefits, inhibition of polyQ protein misfolding and aggregation has received substantial research as the optimal therapeutic strategy for the creation of disease-modifying treatments for polyQ illnesses [20]. In the subsections that follow, we'll concentrate on two strategies for preventing polyQ protein aggregation. One strategy is to create powerful inhibitors, such as small chemical compounds and short peptides, which are intended or chosen to bind specifically to the expanded polyQ tract and prevent polyQ protein aggregation. The alternative strategy is to engage cellular defense mechanisms that stop the formation of aggregates and buildup of misfolded proteins. Both methods effectively slow down the aggregation of enlarged polyQ proteins as well as the course of illness and animal model traits, demonstrating the efficacy of this therapeutic target for drug development [21].

  • In the paragraph 2 "the autophagy dysfunction in polyglutamine diseases" need to be improved.

2.1) the autophagy in HD, SBMA and <SCA-3 might be studied in deep.

Response 2.1:

In response to comment we have include deep study of HD, SBMA and <SCA 3.

Section 4.1 Huntington's disease [In Paragraph 1 & 2]

Most HD patients have a functioning copy of wtHTT because they are heterozygous for the HTT mutation (mHTT). HTT plays a crucial role in maintaining the health of neurons and is crucial for embryonic development. Adult-onset loss of complete HTT causes neurodegeneration. While there have been several attempts to comprehend and create novel treatments that can support pathways that can remove mHTT, and its aggregates, it might be critical to do so selectively without affecting wtHTT levels [26]. Defect in autophagy machinery as well as in UPS system is the hallmark of many neurodegenerative diseases including HD [28]. The polyglutamine tracts causing various toxicities are assumed to hinder the autophagy-related functions of wild type protein in HD [29]. Similarly, the presence of aggregates of mutated N- terminal Htt protein or inclusion bodies (IB) in cell nucleus depicts the alterations in UPS which plays major role in the removal of toxic protein aggregates specifically in nucleus [30]. Various lines of evidence suggest that the pathogenesis of HD is the outcome of defects in autophagy machinery and loss of normal protein function in this cleaning crew and in the UPS system. The formation of autophagosomes and their clearance is normal in HD but these autophagosomes are usually empty due to failure in cargo trapping. The pathophysiology of mutation in several autophagy related genes is similar to pathological findings associated with HD. Studies on loss of wild type protein functions in mouse and Drosophila models revealed several characteristics of disrupted autophagy. Both wild type protein in association with HAP1 and the mutated one can alter the axonal trafficking of autophagosomes and vesicles in neurons in vitro and in vivo [31]. In Drosophila the loss of wild type protein during embryogenesis results in viable embryos but later in adult life absence of it leads to several neurodegenerative symptoms, reduced viability, mobility, reduced or collapsed thorax etc. [32]. Defects in starvation induced autophagy has also been reported in Drosophila larvae and Htt conditional knockout of mouse CNS resulting in accumulation of p62/SQSTM in striatum over time, a hallmark of disturbed autophagy. 

Section 4.2 Spinal and bulbar muscular atrophy [In Paragraph 1]

Pathways that control cellular proteostasis have drawn a lot of interest since the unfolded mutant protein is the primary mediator of toxicity in SBMA. One of them is macroautophagy, often known as "autophagy," a highly conserved catabolic process in which misfolded or defective proteins and organelles in the cytoplasm are encapsulated in double-membraned structures called "autophagosomes."[36]. these autophagosomes are transported to lysosomes and fused with them, allowing their intraluminal contents to degrade. The Coordinated Lysosomal Expression and Regulation (CLEAR) network, which is one of the mechanisms used to regulate autophagy, includes the transcription factor EB (TFEB), which controls the expression of hundreds of genes relevant to lysosomes and autophagy [37]. The main regulator of both autophagy and lysosomal biogenesis is TFEB. Aside from promoting mTOR inhibition and dietary restriction, therapies that cause autophagy also increase TFEB activity [38]. Although there is evidence that SBMA alters the regulation of protein quality, it is unclear how these regulatory transcription factors are impacted by illness [39]. In contrast to the range of 8 to 34 polyQ stretches, there is an accumulation of LC3 puncta in polyQ-AR disease.

Section 4.5 Spinocerebellar ataxia type 3 [ In Paragraph 1]

A number of stresses, including as the presence of cytotoxic protein aggregates, can cause autophagy, which controls nutrient recycling during times of nutritional scarcity and ensures cell survival [52]. Phosphatidylethanolamine (PE) conjugation and autophagosomal anchoring of six closely related ubiquitin-like modifiers from the Atg8 family—LC3A, LC3B, LC3C, GABARAP, GABARAPL1, and GABARAPL2—control the beginning, development, and elongation of the autophagosomal membrane [53]. By directly engaging with these proteins through conserved LC3-interacting areas, LC3s and GABARAPs play key roles in aiding the recruitment of cargo receptors and other autophagy proteins to autophagosomes (LIRs) [54]. The portion of LC3/GABARAP located on the luminal side of autophagic vesicles is destroyed in the lysosomes together with cargo receptors and substrates, whereas LC3/GABARAP located on the cytosolic face is released by Atg4 protease after fusion with lysosomes [55].

In the Drosophila model, genetically inhibiting autophagy can greatly worsen polyQ-neurotoxicity ataxin-3 on retinal cells, pointing to autophagy's protective function in SCA3. By blocking mTOR, the autophagy inducer rapamycin can increase autophagy's activity and encourage the breakdown of ataxin-3 aggregation [58].

2.2) In SCA-7, SCA-17 and DRPLA no description of autophagy has been detailed in the text. Is not autophagy involved in these diseases? Please, describe the autophagy impairment (if observed) in these disease.

Response 2.2:  

Section 4.6 Spinocerebellar ataxia type 7 [In paragraph 1]

However, studies from other polyQ illnesses suggested that polyQ-ATXN7 toxicity might target autophagy. In the context of SCA-7, ATXN7 is a recognized component of the multiprotein SAGA complex, a transcriptional coactivator that has been conserved from yeast to humans. This transcriptional activator (RNAPII) depends on SAGA’s chromatin modelling activity, which directly correlates with the lysosomal membrane degradation (Microautophagy) [61].  

Section 4.7 Spinocerebellar ataxia type 17 [In paragraph 1]

That TBP interacts with other protein factors, including high mobility group box 1 (HMGB1), to regulate gene expression [63]. HMGB1 was found to be sequestered into polyQ-TBP aggregates in vitro, leading to impaired starvation-induced autophagy. HMGB1 is an important autophagy regulator, raising the prospect of a general mechanism for autophagy dysfunction in all polyQ diseases via HMGB1 inhibition [64].

  • In the paragraph 5 "Autophagy modulators as therapeutics against polyQ diseases" the authors described in the text only compounds found effective in HD, but they don't describe compounds for the other polyQ diseases (e.g. Trehalose). Moreover, they list compounds in the table that were not described in the text. Please describe them in the text.

In response to comment we have described compounds effectiveness in HD, SCA1 &6.

Response 3:  

SCA1 & SCA6 exhibits remarkable phenotypic anticipation and diversity, which mostly reflects variations in repeat size across afflicted individuals. The majority of people with SCA1 show indications of extensive brainstem and cerebellar dysfunction with only modest supratentorial involvement. Acetazolamide majorly inhibit the production of bicarbonate ions and dysregulate the pH of the blood in brainstem [86]. In other polyQ disease like SBMA, the male hormones testosterone and dihydrotestosterone (DHT) attach to this dysfunctional receptor, harming the muscle-innervating nerve cells and resulting in weakness. DHT synthesis is reduced by Dutasteride [87].

  • In the text is cited the Protein Quality control system, but it is not described.

In response to comment we have define and describe about Protein Quality Control.

Response 4:

There are two degradation systems reported under the protein quality control system to get rid of these misfolded proteins-the ubiquitin-proteasome system (UPS) or autophagy [22]. In the eukaryotic system, these long polyQ stretches structurally transform into misfolded proteins, which cannot be degraded by UPS machinery, as there is the sequestration of UPS components, due to which delivery of misfolded proteins in polyQ disorders cannot reach out to the nuclear proteasomes [23]. Due to an increase in UPS components and heat shock proteins in nuclear inclusions, these system are deficient, which causes aggravation of polyQ aggregates that become the substrates for autophagy degradation. The autophagy process gets up regulated to keep the protein aggregates for maintaining protein homeostasis. The autophagy process is limited to the cytosol compared to UPS, which is active in both the nucleus and the cytosol.

Minor Comments:

  • The paragraph 3 might be moved before the paragraph 2

Response 1:

  • The English form need to be revised

Response 2:

In response to the comment we have improved certain statement and language in our final corrected manuscript.

Reviewer 2 Report

The manuscript submitted by Mukul Jain et al try to dissect the role of autophagy as disease mechanisms and/or possible therapeutic approach for polyQ diseases. Although the structure of the review is well organized, the content is often partial, incomplete, or repeated. Even simple details such as the use of abbreviations or gene names are often wrong. Trehalose which has been used in in vitro studies and animal models of HD and SBMA is not reported in Figure 3 and Table 1. In the figures, autophagosomes should be represented with a double membrane while mature autophagolysosome and lysosome have a single membrane. In the figures, autophagosomes should be represented with a double membrane while mature lysosomes and autophagolysosomes have a single membrane. Introduction is confused. In chapter 3, the disease descriptions should be better balanced, and the disease described in the same way. The not always precise use of the English language makes reading not always easy and clear.

Few examples:

Lane 29 the use of & is uncommon.

Lane 45 the word acid is missing after amino.

Lane 65 autophagy is a degradative …

Lane 112 aggravation of aggregates, what does it mean?

Lane 115 correlation between aggregate localization and degradative pathway is not shown in figure 2.

Lane 141 what does the abbreviation AV means?

Lane 158 HSP90 instead of Hsp90

Author Response

Response to Reviewer 2

Comments:

The manuscript submitted by Mukul Jain et al try to dissect the role of autophagy as disease mechanisms and/or possible therapeutic approach for polyQ diseases.

  • Although the structure of the review is well organized, the content is often partial, incomplete, or repeated. Even simple details such as the use of abbreviations or gene names are often wrong.

Response: In response to the comment we have completed incomplete sentences with improvised English language & given abbreviations or genes names in revised manuscript.

Ex.,          VPS34 - Phosphatidylinositol 3-Kinase

WDR45 - WD repeat domain 45

LC3 - Light Chain 3

  • Trehalose which has been used in in vitro studies and animal models of HD and SBMA is not reported in Figure 3 and Table 1.

Response: In response to the comment we have reported mode of action of trehalose in HD (mTORC1 inhibition) and SBMA (Decrease the DHT production) in Figure 3 and Table 1.

  • In the figures, auto phagosomes should be represented with a double membrane while mature autophagolysosomes and lysosome have a single membrane. In the figures, autophagosomes should be represented with a double membrane while mature lysosomes and autophagolysosomes have a single membrane. Introduction is confused.

Response: In response to comment we have changed the autophagolysosomes and lysosomes in single membrane structure & we have rearrange introduction portion.

  • In chapter 3, the disease descriptions should be better balanced, and the disease described in the same way. The not always precise use of the English language makes reading not always easy and clear.

Response: In response to comment we have describe in detail about polyQ diseases in revised manuscript, Chapter 3 is updated as Chapter 4 in revised manuscript.

Section 4.1 Huntington's disease [In Paragraph 1 and 2]

Most HD patients have a functioning copy of wtHTT because they are heterozygous for the HTT mutation (mHTT). HTT plays a crucial role in maintaining the health of neurons and is crucial for embryonic development. Adult-onset loss of complete HTT causes neurodegeneration. While there have been several attempts to comprehend and create novel treatments that can support pathways that can remove mHTT, and its aggregates, it might be critical to do so selectively without affecting wtHTT levels [26]. Defect in autophagy machinery as well as in UPS system is the hallmark of many neurodegenerative diseases including HD [28]. The polyglutamine tracts causing various toxicities are assumed to hinder the autophagy-related functions of wild type protein in HD [29]. Similarly, the presence of aggregates of mutated N- terminal Htt protein or inclusion bodies (IB) in cell nucleus depicts the alterations in UPS which plays major role in the removal of toxic protein aggregates specifically in nucleus [30]. Various lines of evidence suggest that the pathogenesis of HD is the outcome of defects in autophagy machinery and loss of normal protein function in this cleaning crew and in the UPS system. The formation of autophagosomes and their clearance is normal in HD but these autophagosomes are usually empty due to failure in cargo trapping. The pathophysiology of mutation in several autophagy related genes is similar to pathological findings associated with HD. Studies on loss of wild type protein functions in mouse and Drosophila models revealed several characteristics of disrupted autophagy. Both wild type protein in association with HAP1 and the mutated one can alter the axonal trafficking of autophagosomes and vesicles in neurons in vitro and in vivo [31]. In Drosophila the loss of wild type protein during embryogenesis results in viable embryos but later in adult life absence of it leads to several neurodegenerative symptoms, reduced viability, mobility, reduced or collapsed thorax etc. [32]. Defects in starvation induced autophagy has also been reported in Drosophila larvae and Htt conditional knockout of mouse CNS resulting in accumulation of p62/SQSTM in striatum over time, a hallmark of disturbed autophagy. 

Section 4.2 Spinal and bulbar muscular atrophy [In Paragraph 1]

Pathways that control cellular proteostasis have drawn a lot of interest since the unfolded mutant protein is the primary mediator of toxicity in SBMA. One of them is macroautophagy, often known as "autophagy," a highly conserved catabolic process in which misfolded or defective proteins and organelles in the cytoplasm are encapsulated in double-membraned structures called "autophagosomes."[36]. these autophagosomes are transported to lysosomes and fused with them, allowing their intraluminal contents to degrade. The Coordinated Lysosomal Expression and Regulation (CLEAR) network, which is one of the mechanisms used to regulate autophagy, includes the transcription factor EB (TFEB), which controls the expression of hundreds of genes relevant to lysosomes and autophagy [37]. The main regulator of both autophagy and lysosomal biogenesis is TFEB. Aside from promoting mTOR inhibition and dietary restriction, therapies that cause autophagy also increase TFEB activity [38]. Although there is evidence that SBMA alters the regulation of protein quality, it is unclear how these regulatory transcription factors are impacted by illness [39]. In contrast to the range of 8 to 34 polyQ stretches, there is an accumulation of LC3 puncta in polyQ-AR disease.

Section 4.5 Spinocerebellar ataxia type 3 [ In Paragraph 1]

A number of stresses, including as the presence of cytotoxic protein aggregates, can cause autophagy, which controls nutrient recycling during times of nutritional scarcity and ensures cell survival [52]. Phosphatidylethanolamine (PE) conjugation and autophagosomal anchoring of six closely related ubiquitin-like modifiers from the Atg8 family—LC3A, LC3B, LC3C, GABARAP, GABARAPL1, and GABARAPL2—control the beginning, development, and elongation of the autophagosomal membrane [53]. By directly engaging with these proteins through conserved LC3-interacting areas, LC3s and GABARAPs play key roles in aiding the recruitment of cargo receptors and other autophagy proteins to autophagosomes (LIRs) [54]. The portion of LC3/GABARAP located on the luminal side of autophagic vesicles is destroyed in the lysosomes together with cargo receptors and substrates, whereas LC3/GABARAP located on the cytosolic face is released by Atg4 protease after fusion with lysosomes [55].

In the Drosophila model, genetically inhibiting autophagy can greatly worsen polyQ-neurotoxicity ataxin-3 on retinal cells, pointing to autophagy's protective function in SCA3. By blocking mTOR, the autophagy inducer rapamycin can increase autophagy's activity and encourage the breakdown of ataxin-3 aggregation [58].

Section 4.6 Spinocerebellar ataxia type 7 [In paragraph 1]

However, studies from other polyQ illnesses suggested that polyQ-ATXN7 toxicity might target autophagy. In the context of SCA-7, ATXN7 is a recognized component of the multiprotein SAGA complex, a transcriptional coactivator that has been conserved from yeast to humans. This transcriptional activator (RNAPII) depends on SAGA’s chromatin modelling activity, which directly correlates with the lysosomal membrane degradation (Microautophagy) [61].  

Section 4.7 Spinocerebellar ataxia type 17 [In paragraph 1]

That TBP interacts with other protein factors, including high mobility group box 1 (HMGB1), to regulate gene expression [63]. HMGB1 was found to be sequestered into polyQ-TBP aggregates in vitro, leading to impaired starvation-induced autophagy. HMGB1 is an important autophagy regulator, raising the prospect of a general mechanism for autophagy dysfunction in all polyQ diseases via HMGB1 inhibition [64].

Section 4.8 Dentatorubral pallidoluysian atrophy [DRPLA] [In paragraph 1]

The fruit fly Drosophila melanogaster is one of the invertebrate models for DRPLA neurodegeneration. When a human disease allele is introduced, it causes age-dependent neuronal death that may be brought on by issues with lysosomal autophagy and accumulation of the poly-Q repeat protein [71].

Round 2

Reviewer 1 Report

The Authors have answered to all the requests.

In my opinion, the manuscript is accepted for publication. 

Author Response

not applicable 

Reviewer 2 Report

Even if the manuscript submitted by Mukul Jain et al has been improved a lot of mistakes and inaccuracies are still present.

The approved name for LC3 is microtubule associated protein 1 light chain 3 (line 254) not only light chain 3 as written in the line 96 

In the same way at line 83 the approved symbol for VPS34 is PIK3C3.

It is also commonly to use abbreviation only to refer to a name previously used in the text (i.e. microtubule associated protein 1 light chain 3 (LC3) not LC3 (light chain 3)).

The approved name of the following abbreviation is missing: mTOR, TORC1, BCL, GABARAP, HSP70, TECPR1, HAP1….

A lot of repetitions are still present and why the authors removed the introduction section? Isn’t it necessary in the structure of the manuscript for biomedicines?

Unfortunately, the mechanism of action of trehalose is not the reduction of DHT but it enhances autophagy and AR degradation in SBMA.

Author Response

Response to Reviewer 2:

Minor comments:

The authors need to carry out the revision as per the suggestions of Reviewer 2 as follows and the editors can go ahead for the reconsideration of the manuscript after the given revision. The approved name for LC3 is microtubule associated protein 1 light chain 3 (line 254) not only light chain 3 as written in the line 96 In the same way at line 83 the approved symbol for VPS34 is PIK3C3. It is also commonly to use abbreviation only to refer to a name previously used in the text (i.e. microtubule associated protein 1 light chain 3 (LC3) not LC3 (light chain 3)). The approved name of the following abbreviation is missing: mTOR, TORC1, BCL, GABARAP, HSP70, TECPR1, HAP1…. A lot of repetitions are still present and why the authors removed the introduction section? Isn’t it necessary in the structure of the manuscript for biomedicines? Unfortunately, the mechanism of action of trehalose is not the reduction of DHT but it enhances autophagy and AR degradation in SBMA.

Response:

According to the minor comment of Reviewer 2, we have symbolize appropriate approved names of abbreviations in middle of the revised manuscript.

As per the comment regarding the removal of introduction section has been added in revised manuscript.

The mode of action of Trehalose in SBMA was correctly edited with reference in Table No.1.